# A systematic review on the influence of coagulopathy and immune activation on New Onset Atrial Fibrillation in patients with sepsis

**Brian W. Johnston**[1,2,3‡]*, **Michael Downes**[1‡], **Angela Hall**[2], **Zachary Thomas**[2],
**Ingeborg D. Welters**[1,2,3]

1 Department of Cardiovascular and Metabolic Medicine, Faculty of Health and Life Sciences, Institute of Life Course and Medical Sciences, University of Liverpool, Liverpool, United Kingdom, 2 Liverpool University Hospitals NHS Foundation Trust, Liverpool, United Kingdom, 3 Liverpool Centre for Cardiovascular Science at University of Liverpool, Liverpool John Moore's University and Liverpool Heart & Chest Hospital, Liverpool, United Kingdom

‡ BWJ and MD share joint first authorship on this work.
* brian.johnston@liverpool.ac.uk

This is a Registered Report and may have an associated publication; please check the article page on the journal site for any related articles.

## Abstract

### Introduction

New Onset Atrial Fibrillation (NOAF) is the most common arrhythmia in intensive care. Complications of NOAF include thromboembolic events such as myocardial infarction and stroke, which contribute to a greater risk of mortality. Inflammatory and coagulation biomarkers in sepsis are thought to be associated with NOAF development. The aim of this systematic review and narrative synthesis is to identify inflammatory and coagulation biomarkers as predisposing risk factors for NOAF in sepsis.

### Methods

Three databases (Medline, Cochrane Library, and Scopus) were searched using a predefined search strategy. Inclusion / exclusion criteria were applied, and quality assessments were performed using the Newcastle Ottawa Scale (NOS).

### Results

We identified 1776 articles; and 12 articles were included in this review. 8 articles were retrospective observational studies and 4 were prospective observational studies. There was considerable heterogeneity between studies regarding outcomes, methodological design, quality, definitions and reported biomarkers of interest. There is evidence that C-reactive protein (CRP) is associated with NOAF, with hazard ratios 3.33 (3.32–3.35) p = 0.001 and odds ratios of 1.011 (1.008–1.014) p<0.001. International Normalised Ratio (INR) and fibrinogen may be associated with NOAF with odds ratios reported as 1.837 (1.270–2.656) p = 0.001 and 1.535(1.232–1.914) p<0.001 respectively.

**Data Availability Statement:** All relevant data are within the manuscript and its Supporting Information files.

**Funding:** The author(s) received no specific funding for this work.

**Competing interests:** The authors have declared that no competing interests exist.

## Conclusion

Further research is required to confirm the association between inflammatory and coagulation biomarkers and the development of NOAF in sepsis. A broader evidence base will guide treatment strategies, improving the standard of care for patients who develop NOAF in sepsis. Furthermore, given the heterogeneity between studies consideration should be given to inclusion of immune biomarkers in future core outcome sets for trials investigating NOAF.

## Introduction

New Onset Atrial fibrillation (NOAF) is the most common arrhythmia in patients admitted to the intensive care unit (ICU). The incidence of NOAF is estimated to be 14% across all ICU admissions but may be as high as 43.9% - 46% in patients with septic shock [1–4].

The development of NOAF in critical illness is associated with several modifiable and non-modifiable risk factors such as obesity and poorly controlled diabetes mellitus, increasing age, male sex and European ancestry, respectively [5]. NOAF is associated with a number of comorbidities such as cardiovascular disease, chronic lung disease, heart failure and stroke [5]. These risk factors are commonly observed in patients that suffer from ambulatory atrial fibrillation [6], however, a number of specific risk factors for NOAF in critical illness have been identified, including vasopressor usage, higher Acute Physiology and Chronic Health Evaluation (APACHE) score, inflammation, oxidative stress and electrolyte imbalances [5, 7]. As a result, the development of NOAF is potentially triggered by the pathophysiology of critical illness and iatrogenic interventions [8].

NOAF has been associated with an increased risk of in-hospital and ICU mortality in patients with sepsis [7, 9]. Patients with NOAF have a tenfold increased risk of readmission with AF compared with patients without the arrhythmia [10]. Furthermore, the large number of patients who develop persistent AF are exposed to greater long-term risks including thromboembolic complications, such as a fivefold increase in stroke risk and a twofold increased risk of myocardial ischemia and infarction [11]. A considerable proportion (20%-34%) of NOAF patients later develop persistent AF and a higher risk for long-term complications [10, 12, 13].

In sepsis, the immune response to infection activates the coagulation cascade; this results in clot formation within the microcirculation, a process which is known as immunothrombosis [14, 15]. Activation of the clotting cascade and immunothrombosis is thought to result in tissue damage and eventually fibrosis of the myocardium, which in turn promotes AF [16, 17]. There is evidence that inflammatory biomarkers in immunothrombosis such as neutrophil extracellular traps (NET), histones and P-selectin predispose individuals to arrhythmias [18–20]. Biomarkers in the inflammatory cascade including C-reactive protein (CRP), Il-6 and TNF-α have been associated with AF [17]; and these inflammatory biomarkers are significantly raised in sepsis. While a large-scale meta-analysis of multiple coagulation factors has established a link between the prothrombotic state and AF in the general population [20], this association has not been demonstrated in critical illness where there is a complex interplay between inflammation and coagulation pathways [18]. A recent study conducted by our group highlighted the association between sepsis induced coagulopathy (SIC) and NOAF in critical care patients [21].

The development of NOAF in sepsis is common and is linked to a wide variety of comorbidities and risk factors. However, evidence for interactions between coagulation and immune

activation and the development of NOAF in patients with sepsis is scarce. Identification of such interactions may offer the potential to prevent, treat and manage NOAF in sepsis, reducing both short and long-term complications. This systematic review investigates the association between coagulopathy and inflammationas predisposing factors for the development of NOAF in patients with sepsis.

## Methods

### Protocol design and registration

The study protocol for this systematic review was reported in accordance with the Preferred Reporting Items for Systematic Reviews and Meta-Analyses—Protocols guideline (PRISMA-P) [22, 23] and published in PLOS One [24].

This protocol was registered in the International Prospective Register of Systematic Reviews (PROSPERO: CRD42022385225), to reduce reporting bias, research duplication, and to increase integrity of our research [25, 26].

### Data sourcing and search strategy

We conducted a systematic review of the literature using a predefined search strategy developed with a specialist healthcare research librarian (AH) (S1 Table). Medline [27], Scopus [28], and Cochrane Library [29] were searched using keywords based upon five key concepts: sepsis, inflammatory markers, coagulopathy, coagulation markers and atrial fibrillation. Medical subject heading (MeSH) terms were used in Medline and Cochrane Library to assist in identifying articles. Searches were formally conducted 8th September 2023.

Search results were exported to EndNote X9 (Clarivate). Following the removal of duplicate articles, the results were screened by two reviewers (MD, BWJ), with a third reviewer (IW) acting to resolve any disagreements. Articles were screened in full text against inclusion and exclusion criteria. We did not limit our search results by date of publication.

### Inclusion and exclusion criteria

We wished to explore the association between coagulopathy and inflammatory biomarkers, and NOAF in patients with sepsis in the hospital setting. Studies that included participants under the age of 18 were excluded, as septic pathophysiology in paediatric patients differs considerably from adults [30, 31]. Studies that listed NOAF as an outcome were included. Due to translation limitations, only English language studies or translated studies were eligible. To provide empirical data, studies included for data extraction had to report on at least one biomarker of coagulation and or inflammation. Pharmaceutical trials were excluded as they report data regarding sepsis and NOAF in the context of a particular drug while mainly focusing on treatment outcomes. Single case reports, case series or expert opinions were excluded for data extraction due to limited quality of evidence. Non-full text and non-published articles were excluded (Table 1).

### Outcome measures

**Primary outcome measure.**

1. Development of NOAF in patients with sepsis.

**Secondary outcome measures.**

1. Observed changes in inflammatory and coagulation biomarkers (S1 Table lists biomarkers).

**Table 1. Inclusion and exclusion criteria for title, abstract and full text screening.**

| Inclusion criteria | Exclusion criteria |
| --- | --- |
| Human studies. | Paediatric populations (<18 years of age). |
| Studies describing NOAF in relation to sepsis*/ critical illness, in the title or abstract. | Animal and in vitro studies. |
| Studies written in or translated to English. | Full text not available in English. |
| Studies reporting coagulation and or inflammation biomarkers with NOAF, in title or abstract (S1 Table). | Pharmaceutical trials or articles that only discuss specific treatments (e.g., betablockers). |
| | Expert opinions or single case reports or case series. |
| | Not available as full text articles (abstract only) or non-published articles. |

*Sepsis has been defined as per the third international consensus definition of sepsis [32].

2. Development of NOAF in critical illness.

3. Development of paroxysmal atrial fibrillation.

4. Development of permanent atrial fibrillation.

5. Mortality.

6. Intensive care unit admission.

7. Length of intensive care unit stay.

8. Length of hospital stay.

9. Development of acute kidney injury.

10. Need for renal replacement therapy.

11. Need for vasopressor therapy.

   **Quality assessment (risk of bias).** Risk of bias was assessed using the Newcastle Ottawa Scale (NOS) for the assessment of non-randomised studies. NOS comprises of three domains which contribute to the overall quality score: selection of groups, comparability, and outcome (cohort studies)/ exposure (case-control). Articles are rated semi- quantitatively between zero to nine stars [33].

   Where applicable the Grading of Recommendations Assessment, Development and Evaluation (GRADE) approach was used in assessing the quality of the evidence. The overall certainty in the evidence was rated as follows: very low (1), low (2), moderate (3) and high (4) [34].

## Data extraction

A Microsoft Excel (Microsoft Corp., Redmond, Washington, USA) custom spread sheet was used for data extraction. Two reviewers (MD and BWJ) extracted data independently. The data sheet was trialled prior to data extraction. Data extracted included: study characteristics, patients characteristics, patient outcomes and all data related to inflammatory and coagulation biomarkers, and NOAF. Regarding patient characteristics, comorbidities were grouped for simplification.

## Data analysis

Due to heterogeneity between the studies regarding their methodological quality and design, study outcomes, study definitions, time points for recording measurements and reported

biomarkers of interest, we were unable to conduct a meta-analysis. As a result, a narrative synthesis was conducted.

# Results

## Search results

Our literature search yielded 1776 articles. After duplicate removal, 1489 articles proceeded to title and abstract screening. Subsequently, 37 articles progressed to full text screening. Applying inclusion exclusion criteria resulted in 12 articles which progressed to data extraction (Fig 1: PRISMA diagram).

## Study characteristics

Of the included studies, four were prospective observational studies [35–38] and eight were retrospective observational studies [39–46]. The combined number of patients for prospective and retrospective study designs was n = 1108 (79–629) and n = 21,065 respectively (Table 2). Our search did not yield any randomised controlled trials (RCTs).

Eight studies were conducted in general ICUs [35–38, 40, 42, 43, 45]. In addition, there were two specialised cardiac ICUs (CICUs) [40, 45], two mixed ICUs [36, 37], one surgical ICU [38], one ICU with medical/surgical/trauma patients [42] and two studies included ICUs specifically for COVID-19 patients [35, 43]. Three studies were conducted at the hospital level [39, 41, 46] and one at a medical centre clinic [44].

There was significant heterogeneity in primary outcomes reported across the studies. Six studies [35, 37, 39, 41, 43, 46] reported the development of NOAF as their primary outcome. Bontekoe et al explored the presence of comorbid AF (alongside Chronic Kidney Disease (CKD)–stage 5) [44] and Hayase et al investigated ICU length of stay (LOS) [36], as primary outcomes. The remaining studies listed differing mortality measures as their primary outcome 90-day [42], 30-day [45], in-hospital [40], ICU mortality [38].

Secondary outcomes included ICU/in-hospital LOS reported by four authors [38–40, 43], 28-day mortality [36, 42], in-hospital mortality [36, 39, 41–43, 46] and ECG monitoring [35, 37, 43, 46]. Individual studies described 90-day mortality [45], a history of sepsis [44], administration of corticosteroids [46], acute kidney injury(AKI) [43] and venous thromboembolism (VTE) [43] as secondary outcomes (Table 2).

Inflammatory and coagulation biomarkers were reported in all studies included, however, there was significant heterogeneity between the biomarkers reported [35–46].

## Patient characteristics

The average age of those who developed NOAF was greater than those who did not, non-NOAF 68.2 ± 3.1 and NOAF: 69.7 ± 3.1 [35]; non-NOAF: 60.7±17.6 and NOAF: 70.1±12.5 [41], mean age in years ± standard deviation.

The average age of the cohorts ranged from 52.5 ± 19.6 [37] to 69.7 ± 3.1 years [35] (S2 Table). All studies reported age, sex, and comorbidities, with the overall male to female ratio being 1.33 (12,335/9,258)(m/f) [35–46]. Eight studies reported results for clinical characteristics, including heart rate and other vital signs [36, 39–43, 45, 46]. Illness severity scores were reported in nine studies [35–42, 45].There was considerable variation regarding reported comorbidities, conditions such as ischaemic heart disease, hypertension, respiratory diseases, and diabetes mellitus were well reported. The APACHE II and IV scores were reported in three studies, APACHE II in two studies [35, 37, 39] and APACHE IV in one study [40]. The Sequential Organ Failure Assessment (SOFA) Score was reported in five studies [35, 36, 38, 39,

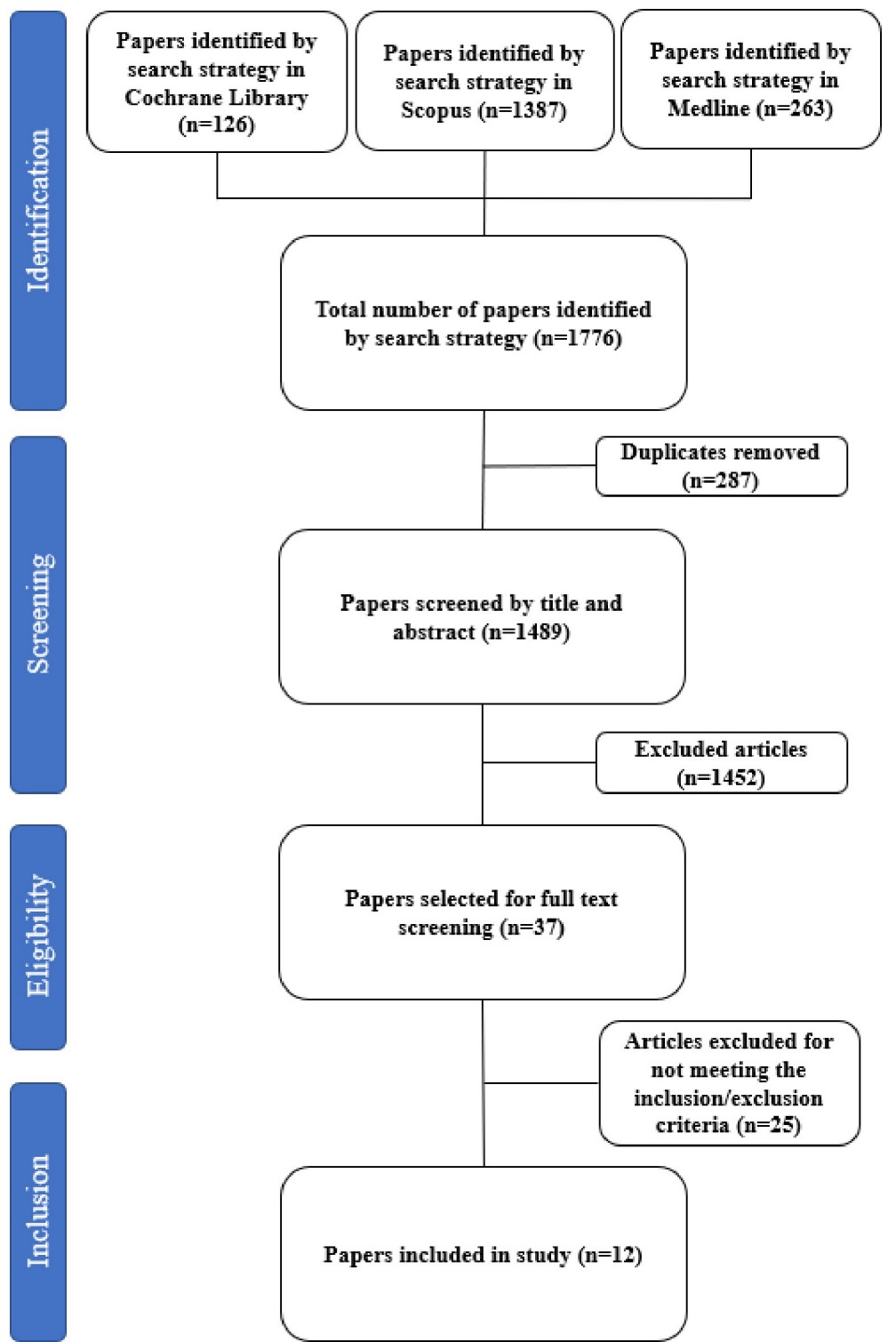

**Fig 1. PRISMA flow diagram displaying the search results using the search strategy described.**

**Table 2. Study characteristics.**

| Study (author and year of publication) | Cohort Size | Inclusion/exclusion criteria | Primary outcomes | Secondary outcomes | Inflammatory and coagulation biomarkers |
|---|---|---|---|---|---|
| **Prospective Observational Studies** | | | | | |
| Zakynthinos, G. E. et al. (2022) | 79 | Inclusion criteria:<br>• Positive for SARS- COV-2<br>• Acute respiratory distress syndrome upon admission.<br>Exclusion criteria:<br>• Cardiac disease<br>• Severe lung disease<br>• Brief AF episodes<br>• NOAF in wards/ICU<br>• Death <48hrs of ICU admission. | Development of NOAF in the ICU and factors leading to its occurrence. | The course of NOAF during ICU stay. Cardiac involvement (ECG and troponin levels monitored) (secondary septic episodes and mortality also reported) | D-dimer, ferritin, WBC, CRP, and troponin. |
| Hayase, N. et al. (2016) | 267 | Inclusion criteria:<br>• Adults >20 years.<br>Exclusion criteria:<br>• Pregnancy<br>• Chronic kidney disease,<br>• Chronic heart disease<br>• Chronic arrhythmia | ICU length of stay. | In-hospital mortality and 28-day overall mortality. | NT-proBNP. |
| Makrygiannis, S. S. et al. (2014) | 133 | Inclusion criteria:<br>• Non-cardiac ICU patients: mechanical ventilation, trauma, and non-cardiac surgery patients.<br>Exclusion criteria:• AF (chronic, intermittent or at admission). | The development of NOAF. | ECG changes and circumstances of NOAF onset. | CRP |
| Meierhenrich, R. et al. (2010) | 629 | Inclusion criteria:<br>• Non-cardiac surgical ICU patients.<br>• Development of NOAF and fulfilled septic shock criteria.<br>Exclusion criteria:<br>• Prior AF. | ICU mortality. | 28-day mortality, 60-day mortality, ICU length of stay. | CRP |
| **Retrospective Observational Studies** | | | | | |
| Li, Z. et al. (2022) | 1568 & 924 | Inclusion criteria:<br>• Adults with sepsis<br>Exclusion criteria:<br>• Incomplete data<br>• Pregnancy<br>• Age<18<br>• Prior heart diseases<br>• Congenital coagulation disorders<br>• Cardiac surgery<br>• Implanted devices<br>• History of AF<br>• Death at ≤24 hours. | The occurrence of NOAF in sepsis. | Length of in-hospital and ICU stay. ICU readmission and in-hospital mortality. | WBC, CRP, procalcitonin, D-dimer, fibrinogen, INR, APTT, BNP, troponin I, platelet count, platelet distribution width. |
| Zhai, G. et al. (2021) | 5512 | Inclusion criteria:<br>• Adults >18 years.<br>• Hospitalized >2 days.<br>Exclusion criteria:<br>• Non-heart disease patients<br>• Missing data<br>• Haematological malignancy. | In-hospital mortality. | Length of CICU stay and length of hospital stay. | MLR (monocyte-lymphocyte ratio) |

*(Continued)*

**Table 2.** (Continued)

| Study (author and year of publication) | Cohort Size | Inclusion/exclusion criteria | Primary outcomes | Secondary outcomes | Inflammatory and coagulation biomarkers |
|---|---|---|---|---|---|
| Ruiz, L. et al.* (2021) | 1092 | Inclusion criteria:<br>• Blood culture result ≤24hrs of admission.<br>Exclusion criteria:<br>• Pneumonia (≤ 3 months)<br>• AF (including paroxysmal). | Occurrence of NOAF (and associated risk factors) | Timing and duration of NOAF, all-cause mortality (in-hospital and 6 months after discharge). | CRP and WBC. (inflammation) |
| Long, Y. et al. (2021) | 7528 | Inclusion criteria:<br>• coagulation function tests ≤24hrs of admission.<br>Exclusion criteria:<br>• age >80 years<br>• Congenital or valvular heart diseases<br>• Pregnancy<br>• Congenital coagulation disorders<br>• Coronary artery stenosis<br>• Implanted cardiac devices.<br>• Admission to cardiac ICU/ cardiac surgery unit. | 90-day mortality. | Morbidity in AF. In-hospital, ICU and 28-day mortality. | PLT, INR, APTT. |
| Kanthasamy, V. et al. (2021) | 109 | Inclusion criteria:<br>• Age >18• positive COVID-19 nasal/throat swabExclusion criteria:<br>• AF (permanent or persistent). | Development of NOAF. | AKI, ECG alterations, VTE (venous thromboembolism), arterial embolism, therapeutic anticoagulation, length of ICU stay, in-hospital death. | WBC, CRP. ferritin, troponin T and D-dimer. |
| Bontekoe, J. et al. (2020) | 97 | Inclusion criteria:<br>• Age >18 years<br>• CKD5 (chronic kidney disease stage 5) on haemodialysis.<br>Exclusion criteria:<br>• Critically ill<br>• Pregnant<br>• Mechanical ventilation | Presence of comorbid AF. | History of sepsis. | PCT, ANG-1, ANG-2, CRP, CD40-L, D-dimer, TNF-α and VWF. |
| Sun, H. et al. (2019) | 3563 | Inclusion criteria:<br>• Age >18<br>• Hospitalised (>2 days)<br>Exclusion criteria:<br>• Haematological diseases<br>• >5% missing data<br>• No neutrophil or lymphocyte data (≤24hrs admission). | 30- day mortality. | 90- day mortality. | NLR, neutrophil count, lymphocyte count, platelet count and WBC. |
| Kindem, Ingvild A. et al. (2008) | 672 | Inclusion criteria:<br>• Positive E. coli or S. pneumoniae.<br>• Paroxysmal AF.<br>Exclusion criteria:<br>• Permanent AF. | Development of AF in a cohort with an existing bacteraemia. | Identification of CRP as a risk factor. Administration of corticosteroids. ECG monitoring, symptoms and in-hospital mortality. | CRP |

*Prospective database utilised. CRP- C-reactive protein. MLR—monocyte-lymphocyte ratio. WBC- white blood cell count. PLT–platelet count. INR–international normalised ratio. APTT- activated partial thromboplastin time. PCT–procalcitonin. ANG-1 –angiopoietin 1. ANG-2 –angiopoietin 2. CD40-L–cluster differentiation of 40. TNF- α –tumour necrosis factor alpha. VWF- Von Willebrand factor. NLR- neutrophil lymphocyte ratio.

42]. The Acute Physiology Score (APS) was reported in one study [40],and two authors reported APSIII [42, 45], the simplified version was reported in three studies (SAPSII) [38, 39, 42]). Pneumonia severity index (PSI) [41], Elixhauser comorbidity index (ECI) and Overall Anxiety Severity and Impairment Scale (OASIS) [42] were used in one study. Due to the variation in illness severity scores used, comparisons of the severity of illness within cohorts were limited (S2 Table).

Imaging results and medications (reported at baseline under demographics), were poorly reported. Four studies reported imaging results with a focus on pulmonary embolism diagnoses [35, 37, 41, 43]. Eight studies reported baseline medication data at baseline [35, 36, 38–41, 45, 46], however, there was a significant variation in the reporting of drugs administered during critical illness, e.g. corticosteroids were only reported in three studies [35, 39, 46] (S3 Table).

Eight studies reported laboratory parameters (including PaO2 etc.) in their baseline characteristics [35, 36, 39–42, 44, 45], however, there was significant variation in the reported parameters. Patient ethnicity data was poorly reported with only three studies providing data [40, 42, 45] (S3 Table).

## Quality assessment results (risk of bias)

Five studies had a NOS score ≤6 and were considered to have low methodological quality [37, 38, 40, 44, 46]. Four studies were of moderate quality with a NOS score of 7 [35, 41, 43, 45], and three studies had little to no risk of bias, scoring 8 points [36, 39, 42] (S4 Table). The Grading of Recommendations Assessment, Development and Evaluation (GRADE) to assess the quality of evidence could not be performed due to insufficient data retrieved regarding the imprecision (focuses on 95% confidence intervals) and inconsistency domains (relies on tests for heterogeneity: $I^2$ and chi-squared).

## Primary outcome

**Development of NOAF in patients with sepsis.** Eight studies reported data on the development of NOAF in patients with sepsis [35–39, 41, 43, 46]. The reported incidence of NOAF varied significantly between 7.8% [38] and 24% [35]. Zakynthinos et al. stated that NOAF occurred during secondary septic episodes in 84.2% of patients [35] (Table 3). There was high variability in patient populations; patients had a wide range of conditions which were not limited to sepsis. Due to this heterogeneity combined with a lack of statistical data (odd ratios and hazard ratios for NOAF in sepsis) no further statements can be made regarding the association between sepsis and NOAF.

## Secondary outcomes

**Development of NOAF in critical illness.** Eight studies included critically ill populations within the intensive care unit (ICU) setting in the context of NOAF [35–38, 40, 42, 43, 45].

**Development of paroxysmal or permanent atrial fibrillation.** Paroxysmal and permanent AF were poorly reported, with only four studies classifying the type of AF [35, 37, 38, 41]. Two studies stated that 42.9% [38] and 50% [37] of patients who converted to sinus rhythm, experienced temporary recurrences of AF. Only Makrygiannis et al provided a percentage of patients who experienced persistent AF (10% [37]). The majority of studies did not provide data for persistent AF (Table 3).

**Mortality.** Mortality was reported in the majority of studies [35, 36, 38–43, 45, 46]. Studies reported differing mortality endpoints (i.e., 28 days, 60 days etc.), limiting the ability to draw comparisons between studies. Patients who developed NOAF had a greater mortality

**Table 3. Reporting of NOAF development as a primary outcome.**

| Study (author and year of publication) | Development of NOAF | Paroxysmal AF | Permanent/ persistent AF |
|---|---|---|---|
| **Prospective Observational Studies** | | | |
| Zakynthinos, G. E. et al. (2022) | 19/105 (24%)* developed NOAF. | 4/105* (3.8%) recurring episodes lasting ≤ 30 mins. | 3/105* (2.9%) sinus rhythm not restored. |
| | NOAF developed 18 ± 4.8 ‡ days after COVID symptoms, or 8.5 ± 2.1 ‡ ICU days (range 3–23 days). | | |
| | 16/19 (84.2%) * secondary sepsis simultaneously with NOAF, | | |
| | 15/19 (81.3%) * septic shock. | | |
| | 17/19 (87.5%) *developed NOAF in first septic episode. | | |
| Hayase, N. et al. (2016) | Developed NOAF: | n/r | n/r |
| | 5/172 (2.9%)* non-sepsis | | |
| | 6/95 (6.3%) * sepsis group | | |
| | OR = 0.56 (-1.29 to 0.18) [a] | | |
| | p = 0.14 | | |
| Makrygiannis, S. S. et al. (2014) | Developed NOAF: | 10/20 (50%) * relapses after cardioversion | 2/20 (10%) * persistent AF. |
| | 20/133 (15%) * total | | |
| | 14/90 (15.6%) * males. | 8/10 (80%) * occurring ≤48hrs of NOAF. | |
| | 6/43 (14.0%) * females | | |
| | 15/71 (21.1%) * medical | | |
| | 2/13 (15.4%) * surgical | | |
| | 3/49 (6.1%) * trauma | | |
| Meierhenrich, R. et al. (2010) | 49/629 (7.8%) * developed NOAF, | 21/49 (42.9%) * temporary recurrence of AF after restoration. | 7/49 (14.3%)* persistent AF. |
| | 38/413(9.2%) * males, | | |
| | 11/216 (5.1%) * females | | |
| | 33/49 (67%) * occurred ≤3 days. | | |
| **Retrospective Observational Studies** | | | |
| Li, Z. et al. (2022) | NOAF occurrence | n/r | n/r |
| | Total = 269/2492 (10.8%) * | | |
| | Training cohort = 167/1568 (10.7%) * | | |
| | Validation cohort = 102/924 (11.0%) * | | |
| | p = 0.763 | | |
| Zhai, G. et al. (2021) | n/r | n/r | n/r |
| Ruiz, L. et al. (2021) | 109/1092 (9.9%) * total | 78/109(71.6%) *; 64/87(73.6%) * ER & 14/22(63.6%) * in- hospital. | 31/109(28.4%)* (23/87(26.4%)*ER & 8/22(36.4%)* in-hospital |
| | 87/109 (79.8%) * on ER admission. | | |
| | 22/109 (20.2%) * developed NOAF during hospitalisation | | |
| Long, Y. et al. (2021) | n/r | n/r | n/r |
| Kanthasamy, V. et al. (2021) | 16/109 (14.6%) * with NOAF | n/r | n/r |
| Bontekoe, J. et al. (2020) | n/r | n/r | n/r |
| Sun, H. et al. (2019) | n/r | n/r | n/r |
| Kindem, Ingvild A. et al. (2008) | Developed NOAF: | n/r | n/r |
| | 104/672 (15.4%) * total | | |
| | 61/ 384 (15.9%) * women | | |
| | 43/ 288 (14.9%) *men | | |

*n (%) ‡ Mean ± standard deviation (SD) † Median (Interquartile range)

[a] Odds Ratio/Hazard ratio/Regression (confidence interval 95%)

compared to those who did not [35, 38–43, 45, 46]. Ruiz et al reported that the mortality rate for patients with NOAF was 17.9% vs 2.9% in the non-NOAF group, p<0.001 [41]. Kanthasamy et al determined an OR for mortality in patients with NOAF of 5.4; (95% CI: 1.7–17); p = 0.004 [43] (S5 Table).

**ICU admission, hospital length of stay and ICU length of stay.**   ICU admission data, LOS in-hospital, LOS ICU were poorly reported. Only Long et al provided an OR for AF by ICU type [42], and Li et al provided ICU re-admission data [39]. Seven of the studies did not report LOS data [35–37, 42, 44–46] (S5 Table).

**Development of Acute Kidney Injury (AKI) and need for Renal Replacement Therapy (RRT).**   Data for AKI and RRT were infrequently reported. Two studies reported renal disease without specifying whether this was acute or chronic [41, 42] (S5 Table)). Kanthasamy et al reported AKIs requiring RRT, 64% of patients had AKI, of which 36% required RRT. Importantly, AKIs were more common in patients with NOAF 94% vs 54% in patients without NOAF, p = 0.028 [43].

**Need for vasopressor therapy.**   Vasopressor usage was reported in four studies [35, 36, 38, 39]. There was no consensus regarding the reporting of drugs administered, with one study reporting noradrenaline usage [35] and two studies reporting dobutamine usage [36, 38]. Li et al reported that NOAF was more common in patients treated with dopamine, OR: 1.876 (95% CI: 1.227–2.874), p = 0.004 [39] (S6 Table).

**Inflammatory and coagulation biomarkers and the development of NOAF.**   Inflammatory and coagulation biomarkers were poorly reported across all studies (S7 Table). Six studies which reported the development of NOAF as their primary outcome [35, 37, 39, 41, 43, 46], also reported results for inflammation and coagulation biomarkers (Table 4).

## C-reactive protein

Six studies reported CRP data in relation to the development of NOAF [35, 37, 39, 41, 43, 46]. Two studies described a rise in CRP prior to NOAF onset: 3 days prior = 7.41 ± 4.3 mg/dL vs

**Table 4. Inflammatory and coagulation biomarkers for studies with NOAF as the primary outcome.**

| Study (author and year of publication) | Inflammatory and coagulation biomarkers | | | | | | | |
|---|---|---|---|---|---|---|---|---|
| | CRP | WBC | D-Dimer | Ferritin | Troponin I | Troponin T | INR | Fibrinogen |
| **Prospective Observational Studies** | | | | | | | | |
| Zakynthinos, G. E. et al. (2022) | CRP (mg/dl) (< 0.5) | WBC, 109/L | n/r | Ferritin (ng/ml) | Troponin I, ng/ml | n/r | n/r | n/r |
| | 3 days before NOAF | (< 10 ×109/L) | | (< 330) | (< 0.02) | | | |
| | 7.41 ± 4.3 ‡ | 3 days before NOAF | | 3 days before NOAF | 3 days before NOAF | | | |
| | day of occurrence 12.33 ± 4.1 ‡ | | | | | | | |
| | p = 0.01 | 8.680 ± 2.679 ‡ | | 1188 ± 453 ‡ | 0.16 ± 0.31 ‡ | | | |
| | | day of occurrence | | day of occurrence | day of occurrence | | | |
| | | 1.0627 ± 1.972 ‡ | | 999 ± 787 ‡ | 0.64 ± 1.04 ‡ | | | |
| | | p = 0.71 | | p = 0.46 | p = 0.017 | | | |
| | | | | **ORs: NOAF occurrence** | | | | |
| | | | | Ferritin levels | | | | |
| | | | | 1.00 (1.00–1.00) [a] | | | | |
| | | | | p = 0.502 | | | | |

*(Continued)*

**Table 4.** (*Continued*)

| Study (author and year of publication) | Inflammatory and coagulation biomarkers | | | | | | | |
|---|---|---|---|---|---|---|---|---|
| | CRP | WBC | D-Dimer | Ferritin | Troponin I | Troponin T | INR | Fibrinogen |
| Makrygiannis, S. S. et al. (2014) | **CRP mean** **NOAF group** 3 days before = 62.4 (±47.5) [i] mg/L day of onset = 115.0 (±82.1) [i] mg/L 2 days after = 111.4 (±68.0) [i] p = 0.005 | n/r | n/r | n/r | n/r | n/r | n/r | n/r |
| **Retrospective Observational Studies** | | | | | | | | |
| Li, Z. et al. (2022) | **CRP** **OR (CI) for NOAF** 1.011 (1.008–1.014) p<0.001 [a] | n/r | n/r | n/r | n/r | n/r | **INR** **OR (CI) for NOAF** 1.837 (1.270–2.656) p = 0.001 [a] | **Fibrinogen** **OR(CI) for NOAF** 1.535(1.232–1.914) [a] p<0.001 |
| Ruiz, L. et al. (2021) | **HRs for NOAF Inflammation* Mild:** Ref [a] **Moderate:** 1.31 (0.66–2.62) [a] (overall unadjusted) 1.88 (1.87–1.89) [a] (overall adjusted) p = 0.107 1.38 (0.64–2.97) [a] (ER unadjusted) 1.96 (1.95–1.97) [a] (ER adjusted) p = 0.114 **Severe:** 2.15 (1.18–3.89) [a] (overall unadjusted) 2.88 (2.87–2.90) [a] (overall adjusted) p = 0.002 2.16 (1.11–4.21) [a] (ER unadjusted) 3.33 (3.32–3.35) [a] (ER adjusted) p = 0.001 ***inflammation includes CRP and WBC** | **WBC < 4000 (x10^9 /L)** 3.06 (1.54–6.05) [a] (overall unadjusted) 2.65 (2.64–2.67) [a] (overall adjusted) p = 0.027 2.85 (1.33–6.12) [a] (ER unadjusted) 3.42 (3.40–3.44) [a] (ER adjusted) p = 0.008 | n/r | n/r | n/r | n/r | n/r | n/r |

(*Continued*)

**Table 4.** (Continued)

| Study (author and year of publication) | Inflammatory and coagulation biomarkers | | | | | | | |
|---|---|---|---|---|---|---|---|---|
| | CRP | WBC | D-Dimer | Ferritin | Troponin I | Troponin T | INR | Fibrinogen |
| Kanthasamy, V. et al. (2021) | CRP mg/l | WBC (10^9/L) | D-Dimer mg/l | Ferritin µg/l | n/r | Troponin T ng/l | n/r | n/r |
| | 333(256–388) † total | | | | | | | |
| | 324(248–400) † NOAF | 18.0(14.0–25.5) † total | 17(6–59) †total | 1870(1078–2904) † total | | 80(35–160) † total | | |
| | 333(256–388) † non NOAF | 18(13.0–30.0) †NOAF | 18(8–46) †NOAF | 1670(1128–2329) † NOAF | | 83(53–158) †NOAF | | |
| | p = 0.8 | 18(14.0–24.0) † non NOAF | 17(6–60) †non NOAF | 1926(1031–3018) † non-NOAF | | 80(33–160) † non-NOAF | | |
| | | p = 0.73 | p = 0.9 | p = 0.45 | | p = 0.5 | | |
| Kindem, Ingvild A. et al. (2008) | OR(CI) for AF | n/r | n/r | n/r | n/r | n/r | n/r | n/r |
| | CRP1 0–50 mg/l | | | | | | | |
| | 1 (crude) p = 0.27 | | | | | | | |
| | 1.00(adjusted)p = 0.11 | | | | | | | |
| | CRP1 51–150 mg/l | | | | | | | |
| | 1.34 (0.68–2.67) [a] | | | | | | | |
| | p = 0.27 | | | | | | | |
| | 1.49 (0.67–3.32) [a] | | | | | | | |
| | p = 0.11 | | | | | | | |
| | CRP1 151–250 mg/l | | | | | | | |
| | 1.28 (0.65–2.52) [a] | | | | | | | |
| | p = 0.27 | | | | | | | |
| | 1.53 (0.69–3.39) [a] | | | | | | | |
| | p = 0.11 | | | | | | | |
| | CRP1 >250 mg/l | | | | | | | |
| | 1.47 (0.79–2.75) [a] | | | | | | | |
| | 1.94 (0.90–4.16) [a] | | | | | | | |
| | p = 0.11 | | | | | | | |

*n (%) ‡ Mean ± standard deviation (SD) † Median (Interquartile range)

[a] Odds Ratio/Hazard ratio/Regression (confidence interval 95%) [i] Mean ± Standard Error of the mean (SEM). CRP–C-reactive protein. INR- international normalised ratio. WBC–white blood cell count.

day of NOAF occurrence = 12.33 ± 4.1 mg/dL, (mean ± SD) p = 0.01 [35], and 3 days before = 62.4 ±47.5 mg/L vs day of onset = 115.0 ± 82.1 mg/L, (mean ± SEM) p = 0.005 [37]. Two studies provided ORs for NOAF: 1.011 (95% CI: 1.008–1.014), p<0.001 [39] and 1.94 (95% CI: 0.90–4.16) (for CRP>250mg/L) (not significant p = 0.11) [46]. Hazard ratios were provided by Ruiz et al for inflammation and NOAF: 3.33 (3.32–3.35) in severe inflammation, adjusted HR for NOAF in the emergency room, p = 0.001 [41]. In this study CRP and white blood cell count (WBC) were combined and patients with a CRP >150mg/L or a WBC > $30 \times 10^9$/L were classified as having "severe inflammation" [41]. Kanthasamy et al provided median (IQR) CRP values for NOAF and non-NOAF patients which showed no significant difference between the two groups, p = 0.8 [43].

## White blood cell count

Three studies reported WBC data [35, 41, 43]. Similar to the rise in CRP, Zakynthinos et al described a rise in WBC before NOAF development: 3 days before NOAF = 8.680 ± 2.679 vs

day of NOAF occurrence = 1.0627 ± 1.972, WBC<10 x10$^9$/L, (mean ± SD) p = 0.71 [35]. A significant OR was provided for leukocytopenia in the emergency room: 3.42 (3.40–3.44) (adjusted), p = 0.008 [41], showing that patients with a WBC<4 x10$^9$/L. were at greater risk of developing NOAF on admission. Kanthasamy et al reported median (IQR) WBC values, but could not identify significant differences between patients who developed NOAF and those who did not, p = 0.73 [43].

### D-Dimer & ferritin

D-Dimer was poorly reported across all studies. Kanthasamy et al reported D- dimer results with no significant difference between patients developing NOAF versus those who did not (p = 0.9) [43]. Ferritin was poorly reported with only two studies providing data [35, 43]. Median (IQR) data (ng/ml) were provided: 1670(1128–2329) in NOAF vs 1926(1031–3018) in non-NOAF patients, p = 0.45 [43], a lower median ferritin was observed in the NOAF group. Zakynthinos et al reported ferritin levels for patients who developed NOAF: 3 days before NOAF = 1188 ± 453 vs day of occurrence = 999 ± 787, (ng/mL)(mean ± SD) p = 0.46 [35].

### Troponin I & T

Two studies reported on troponin I [35] & T [43]. There was no significant difference between the NOAF and non-NOAF groups for reported troponin T levels, p = 0.5 [43]. Zakynthinos et al reported a significant increase in troponin I, 3 days before NOAF = 0.16 ± 0.31, compared to the day of occurrence = 0.64 ± 1.04 (ng/ml)(mean ± SD), p = 0.017 [35]. Zakynthinos et al did not specify whether the blood samples were taken before or after the day of NOAF occurrence, this may explain this rise in troponin.

### International normalized ratio (INR) & fibrinogen

INR and fibrinogen were rarely reported, however Li et al found that both INR and fibrinogen were predictive of NOAF following multivariable logistic regression with an OR of 1.837 (95% CI: 1.270–2.656), p = 0.001, for INR and of 1.535(95% CI: 1.232–1.914), p<0.001 for fibrinogen [39]. This indicates that both INR and fibrinogen might be independent predictors for the development of NOAF. Li et al developed a risk prediction score for NOAF that included INR and Fibrinogen in addition to age, congestive heart failure, sequential organ failure assessment score, CRP and Dopamine use.

## Discussion

Age is the strongest risk factor for AF in the general population and for NOAF in critical illness [5, 47–50]. The studies identified in our search support a strong association between age and the risk of developing NOAF [36, 37, 39–43]. The development of NOAF with increasing age has been well described although its pathogenesis is yet to be fully understood. Fibrosis, a key substrate of AF, increases with age along with a loss of cardiac myocytes [51]. In addition to structural remodeling, electrical changes seen with age, such as alterations to the shape and duration of the myocyte action potential [51] contribute to the risk of NOAF in sepsis for older patients.

There is limited evidence that CRP can be used to predict NOAF. Rising CRP levels are associated with an increased risk of NOAF as seen in studies from Zakynthinos et al and Makrygiannis et al [35, 37, 39, 52–54]. Although this association has been demonstrated [39] in this review, it is likely that one marker on its own cannot accurately predict NOAFdue to the myriad of immune mediators and pathways involved in NOAF pathogenesis [6, 55]. In a previous

study, patients with higher CRP levels due to genetic variation were not at an increased risk of developing AF. Thus, high CRP levels may be associated with but not necessarily causative for the development of NOAF [56]. While mainly regarded as a marker of severity of inflammation, it has been postulated that high CRP levels increase the L-type calcium currents that may lead to calcium overload in cardiac myocytes, which in turn can cause diastolic releases of calcium from the sarcoplasmic reticulum to contribute to arrhythmias [57, 58]. However, it is unclear whether high CRP concentrations lead to an increased risk of NOAF, or if NOAF can precipitate an increase in CRP levels. Kallergis et al suggests that AF itself induces inflammation, which in turn raises CRP levels and revealed that in patients who converted into sinus rhythm, a gradual decrease in CRP levels has been observed following cardioversion [59]. In a different study, the sole predictor of elevated CRP was the occurrence of AF; patients with paroxysmal AF had raised CRP levels only during episodes of AF [60].

INR has been identified as an independent predictive factor of NOAF in patients with sepsis and was a key variable in a predictive model for NOAF development [39]. INR is raised in patients with sepsis-induced coagulopathy (SIC) and disseminated intravascular coagulation (DIC) [61] and early coagulation disorder in the first 24hrs after admission was an independent risk factor for AF, with INR and activated partial thromboplastin time (APTT) as predictive markers [42]. Coagulation abnormalities, microthrombosis and activation of the immune system play a role in the development of NOAF in sepsis [20, 39, 42].

Patients with sepsis display significantly higher levels of fibrinogen, increasing their risk of developing NOAF as highlighted by Li et al and described in this review [39]. A meta-analysis of multiple coagulation factors revealed that higher levels of fibrinogen were associated with ambulatory AF [20]. In a prospective study of individuals without cardiovascular diseases, higher levels of fibrinogen also incurred a greater risk of AF [62], supporting the importance of studying coagulation abnormalities in the development of NOAF.

Vasopressors in critical illness have been associated with the development of NOAF [5, 39] and previous studies have shown that patients treated with dobutamine [63] and dopamine [39, 64, 65] were more likely to develop NOAF during treatment for sepsis. However, increased vasopressor usage may be a surrogate marker of disease severity in ICU patients [10]. Zakynthinos et al described a rise in noradrenaline dosage prior to NOAF onset and Li et al described an association between dopamine and NOAF. Taken together, we conclude that vasopressor usage may be associated with the development of NOAF, as Zakynthinos et al outlined a rise in noradrenaline usage prior to NOAF onset, and Li et al reported an OR showing an association between dopamine and NOAF [35, 39]. However, given the scarce evidence available, we cannot be certain of the strength of this association.

This systematic review shows that mortality rates within the NOAF groups were greater than the mortality rates observed in the non-NOAF groups [35, 38–43, 45, 46]. NOAF in sepsis has been strongly linked to both an increase in ICU and in-hospital mortality [7, 8]. Patients with NOAF are at an increased risk of stroke, with potentially devastating effects on quality of life [9, 66]. However, it remains unclear to which degree NOAF represents an indicator of greater illness severity, which may explain the higher overall mortality rates observed in patients with NOAF.

## Strengths and limitations

This systematic review is an up-to-date assessment of the literature regarding the influence of coagulation and immune activation, and the development of NOAF in patients with sepsis. This review covers a broad range of evidence from prospective and retrospective observational studies and adds novel information regarding potential risk factors for NOAF in this particular

patient group. We have used the international consensus definition of sepsis [32] to define our sepsis population. Some studies predate this definition however, it is sufficiently broad to include all relevant studies.

There was considerable variation in the methodological quality (risk of bias) of the studies assessed with the NOS scoring tool, limiting the conclusions we were able to draw from results in the studies. The Grading of Recommendations Assessment, Development and Evaluation (GRADE) could not be performed due to a lack of data regarding the imprecision and inconsistency domains. Our review is limited by a lack of high quality RCT studies which would have increased the overall certainty of our conclusions. There was a lack of statistical data and considerable heterogeneity between studies; therefore we were unable to pool data and conduct a meta-analysis.

## Conclusion

We have identified and described the association between multiple coagulation and inflammatory biomarkers, and the development of NOAF in this review. We have highlighted that CRP, INR, and fibrinogen may be associated with the development of NOAF. In line with previous reports, we could demonstrate that the mortality rates within groups with NOAF were higher than those without, and that increased vasopressor use may also be associated with NOAF development. However, the evidence base is weak due to the heterogeneity between studies regarding outcomes, definitions and reported biomarkers of interest.

In patients with sepsis, other factors including age and vasopressor usage may have more influence on the development of NOAF than inflammatory and coagulation biomarkers. There is a need for further research regarding inflammatory and coagulation biomarkers in large studies to determine the pathogenesis of NOAF. Studies investigating NOAF suffer from significant heterogeneity with regards to the definition of NOAF, its treatments and the outcomes measured. Overall, the reporting of inflammatory and coagulation biomarkers remains poor. Reporting of biomarkers, particularly those used in routine clinical care may yield important information for prediction of NOAF, development of risk prediction tools and prediction of outcomes. We suggest that a core set of outcomes is required for studies into NOAF to allow comparisons between studies, and a set of defined routinely used inflammatory and coagulation biomarkers should be included. A broader evidence base will help to define preventative and therapeutic strategies to improve standard care for patients who develop NOAF during sepsis.

## Supporting information

**S1 Table. Complete search strategy by database.**
(DOCX)

**S2 Table. Patient characteristics.**
(DOCX)

**S3 Table. Additional patient characteristics: Ethnicity, imaging, medications and laboratory parameters.**
(DOCX)

**S4 Table. Quality of methodology (risk of bias).**
(DOCX)

**S5 Table. Additional patient outcomes.**
(DOCX)

**S6 Table. Reporting of vasopressor usage (secondary outcome).**
(DOCX)

**S7 Table. Inflammatory and coagulation biomarkers.**
(DOCX)

## Author Contributions

**Conceptualization:** Michael Downes, Ingeborg D. Welters.

**Data curation:** Michael Downes, Angela Hall.

**Methodology:** Michael Downes, Angela Hall, Ingeborg D. Welters.

**Supervision:** Brian W. Johnston, Ingeborg D. Welters.

**Writing – original draft:** Michael Downes, Zachary Thomas.

**Writing – review & editing:** Brian W. Johnston, Michael Downes, Angela Hall, Zachary Thomas, Ingeborg D. Welters.

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
