## [Decision Letter · Decision Letter 0]

20 Nov 2024

PONE-D-24-15357A Systematic Review on the Influence of Coagulopathy and Immune activation on New Onset Atrial Fibrillation in Patients with SepsisPLOS ONE

Dear Dr. Johnston,

Thank you for submitting your manuscript to PLOS ONE. After careful consideration, we feel that it has merit but does not fully meet PLOS ONE’s publication criteria as it currently stands. Therefore, we invite you to submit a revised version of the manuscript that addresses the points raised during the review process.

We look forward to receiving your revised manuscript.

Kind regards,

Chiara Lazzeri

Academic Editor

PLOS ONE

Journal Requirements:

2. We note that your Data Availability Statement is currently as follows: “All relevant data are within the manuscript and in Supporting Information files.”

Reviewers' comments:

Reviewer's Responses to Questions

**Comments to the Author**

1. Does the manuscript adhere to the experimental procedures and analyses described in the Registered Report Protocol?

If the manuscript reports any deviations from the planned experimental procedures and analyses, those must be reasonable and adequately justified.

Reviewer #1: Yes

Reviewer #2: Yes

2. If the manuscript reports exploratory analyses or experimental procedures not outlined in the original Registered Report Protocol, are these reasonable, justified and methodologically sound?

A Registered Report may include valid exploratory analyses not previously outlined in the Registered Report Protocol, as long as they are described as such.

Reviewer #1: Yes

Reviewer #2: Partly

3. Are the conclusions supported by the data and do they address the research question presented in the Registered Report Protocol?

The manuscript must describe a technically sound piece of scientific research with data that supports the conclusions. The conclusions must be drawn appropriately based on the research question(s) outlined in the Registered Report Protocol and on the data presented.

Reviewer #1: Partly

Reviewer #2: Partly

4. Have the authors made all data underlying the findings in their manuscript fully available?

Reviewer #1: No

Reviewer #2: Yes

5. Is the manuscript presented in an intelligible fashion and written in standard English?

Reviewer #1: Yes

Reviewer #2: Yes

6. Review Comments to the Author

Please use the space provided to explain your answers to the questions above. (Please upload your review as an attachment if it exceeds 20,000 characters)

Reviewer #1: Thank you for the opportunity to review this article entitled “A Systematic Review on the Influence of Coagulopathy and Immune activation on New Onset Atrial Fibrillation in Patients with Sepsis”.

The methods are well specified, and a protocol was published in advance. The search strategy has benefited from the inclusion of a research librarian.

While the number of articles included is relatively small, limiting interpretation, this is an interesting and well-conducted review.

However, there are some changes which should be made prior to consideration for publication to clarify the message, and polish some details:

Abstract

1) In the abstract, it is stated that “a core set of study outcomes should be developed”. This needs to be made more specific. Do the authors mean a core outcome set for studies specifically investigating the association of immune biomarkers with NOAF? Or a core outcome set for studies of NOAF treatment? “including a list of routinely used inflammatory and coagulation biomarkers.” — are the biomarkers an outcome in this context? Or does this refer to a core dataset of potential NOAF predictors to be collected in future studies of NOAF in sepsis?

Methods

2) The authors state “Participants under the age of 18 were excluded”. Was the analysis conducted at the patient level, or would it be more accurate to say that studies of patients under the age of 18 were excluded?

3) Please could the authors add the range (smallest and largest) of patient numbers (study sizes) for prospective and retrospective studies in the “Study characteristics” paragraph.

4) CKD + CRP acronyms undefined in main text.

Results

5) Suggest the authors clarify that the reported OR for NOAF of 1.011 is per unit increase (if this is the case).

6) Possible error and some unnecessary parentheses when reporting units, e.g. “with a WBC<4000 (x109/L) were at…”. This would look better as “with a WBC <4000 ×109/L were at…”. Are these the correct units? I would expect leukocytopenia to be around <4.0 ×109/L, not 4000. Prior to this there is a duplication of units in “WBC x109/L (< 10 ×109/L),”

7) For INR and fibrinogen, the authors should clarify what unit increase / threshold the odds ratios refer to, as they have done for CRP (is this per unit increase?)

Conclusion

8) The authors should clarify whether these biomarkers should form part of a core outcome set, or whether these data would also be useful prior to AF onset (as potential predictors). Would the authors advocate for a core dataset or variable set? Outcomes may only be collected after the event.

Minor

I’m sure the editors will sort this out but hyphens have been used to express ranges where a dash or “to” would be more appropriate.

I’d lose the comma in “… histones and P-selectin, predispose individuals to arrhythmias”.

Same for “A recent study conducted by two of our authors, highlighted the association…”.

And again for “myocyte action potential (51), contribute to the risk of NOAF”.

Discussion – I think “studies identified” would be better than “studies retrieved”.

Discussion – typo “NAOF”.

Reviewer #2: The manuscript titled "A Systematic Review on the Influence of Coagulopathy and Immune Activation on New Onset Atrial Fibrillation in Patients with Sepsis" presents a comprehensive analysis of the relationship between inflammatory and coagulation biomarkers and the development of new onset atrial fibrillation (NOAF) in patients experiencing sepsis.

1- How do the authors propose to standardize the reporting of inflammatory and coagulation biomarkers in future studies to enhance comparability?

2- What specific mechanisms do the authors suggest might link elevated levels of CRP, INR, and fibrinogen to the development of NOAF in septic patients?

3- In light of the findings, what recommendations do the authors provide for clinical practice regarding the monitoring and management of patients with both sepsis and NOAF?

4- The language and grammar of the manuscript could be improved to enhance the clarity and readability of the text. For instance, in the abstract of this manuscript:

... Medline, Cochrane Library and Scopus ... --- > ... Medline, Cochrane Library, and Scopus ...

7. PLOS authors have the option to publish the peer review history of their article (what does this mean?). If published, this will include your full peer review and any attached files.

Reviewer #1: No

Reviewer #2: No

---

## [Author Response · Author response to Decision Letter 0]

7 Jan 2025

We thank the editors and peer reviewers for their comments. Please find below answers to their comments and a revised manuscript with tracked changes. 

Please use the space provided to explain your answers to the questions above. (Please upload your review as an attachment if it exceeds 20,000 characters)

Reviewer #1: Thank you for the opportunity to review this article entitled “A Systematic Review on the Influence of Coagulopathy and Immune activation on New Onset Atrial Fibrillation in Patients with Sepsis”.

The methods are well specified, and a protocol was published in advance. The search strategy has benefited from the inclusion of a research librarian.

While the number of articles included is relatively small, limiting interpretation, this is an interesting and well-conducted review.

However, there are some changes which should be made prior to consideration for publication to clarify the message, and polish some details:

Abstract

1) In the abstract, it is stated that “a core set of study outcomes should be developed”. This needs to be made more specific. Do the authors mean a core outcome set for studies specifically investigating the association of immune biomarkers with NOAF? Or a core outcome set for studies of NOAF treatment? “including a list of routinely used inflammatory and coagulation biomarkers.” — are the biomarkers an outcome in this context? Or does this refer to a core dataset of potential NOAF predictors to be collected in future studies of NOAF in sepsis?

We thank the reviewers for this comment and have altered the abstract to better reflect the intentions of the authors. Future studies should consider reporting immune and inflammatory biomarkers as standard and this may best be realized by their inclusion in a core outcome set for trials investigating NOAF. Line 43-45

Methods

2) The authors state “Participants under the age of 18 were excluded”. Was the analysis conducted at the patient level, or would it be more accurate to say that studies of patients under the age of 18 were excluded?

We thank the reviewer for the above comment and agree. We have revised the manuscript to reflect that we excluded studies of patients under the age of 18 rather than conduct a patient level review. Line 102 and 103

3) Please could the authors add the range (smallest and largest) of patient numbers (study sizes) for prospective and retrospective studies in the “Study characteristics” paragraph.

The manuscript has been updated to include min and max patient numbers). Line 154

4) CKD + CRP acronyms undefined in main text.

The manuscript has been updated to define CKD and CRP respectively. 

Results

5) Suggest the authors clarify that the reported OR for NOAF of 1.011 is per unit increase (if this is the case).

We have updated the manuscript to clarify that this OR is following multivariable logistic regression analysis. 

6) Possible error and some unnecessary parentheses when reporting units, e.g. “with a WBC<4000 (x109/L) were at…”. This would look better as “with a WBC <4000 ×109/L were at…”. Are these the correct units? I would expect leukocytopenia to be around <4.0 ×109/L, not 4000. Prior to this there is a duplication of units in “WBC x109/L (< 10 ×109/L),”

Thank you for highlighting these typographic errors. We have corrected the units and deleted any duplications. 

7) For INR and fibrinogen, the authors should clarify what unit increase / threshold the odds ratios refer to, as they have done for CRP (is this per unit increase?)

Li et al undertook multivariable logistic modelling and provided OR for the significance of different variables. We have updated the manuscript to make this clear in the results section. 

Conclusion

8) The authors should clarify whether these biomarkers should form part of a core outcome set, or whether these data would also be useful prior to AF onset (as potential predictors). Would the authors advocate for a core dataset or variable set? Outcomes may only be collected after the event.

We have added an additional sentence as part of the conclusion. The literature on NOAF suffered from significant heterogeneity. There is a lack of agreement on the definition of NOAF, treatments used and the outcomes used in NOAF studies. As such combining results is problematic. Reporting of inflammatory and coagulation biomarkers whilst promising remains quite poor. We therefore suggest that a core outcome for trials investigating NOAF should be developed. Furthermore, inflammatory and coagulation biomarkers should be included in this core outcome set. Not only to further improve prediction of NOAF and help develop algorithms/tools for prediction but also to provide mechanistic insights into the pathophysiology of NOAF.

Minor

I’m sure the editors will sort this out but hyphens have been used to express ranges where a dash or “to” would be more appropriate.

Thank you for the comment. The authors have reviewed the manuscript and rectified these issues. 

I’d lose the comma in “… histones and P-selectin, predispose individuals to arrhythmias”.

Same for “A recent study conducted by two of our authors, highlighted the association…”.

And again for “myocyte action potential (51), contribute to the risk of NOAF”.

Thank you we have updated the manuscript as per recommendation. 

Discussion – I think “studies identified” would be better than “studies retrieved”.

We have updated the manuscript to include your recommendation.

Discussion – typo “NAOF”.

This has been rectified.

Reviewer #2: The manuscript titled "A Systematic Review on the Influence of Coagulopathy and Immune Activation on New Onset Atrial Fibrillation in Patients with Sepsis" presents a comprehensive analysis of the relationship between inflammatory and coagulation biomarkers and the development of new onset atrial fibrillation (NOAF) in patients experiencing sepsis.

1- How do the authors propose to standardize the reporting of inflammatory and coagulation biomarkers in future studies to enhance comparability?

As mentioned, we advocate for the development of a core outcome set for clinical trials investigating the development of NOAF. As part of the process inflammatory and coagulation biomarkers could be included which would produce a standardized ‘bank’ or ‘panel’ of biomarkers to be reported. However given that this is an advancing field it is important that authors and researchers are not limited to only those biomarkers. A core outcome set would be ideal in this respect as it defines a minimum set of outcomes, whilst allowing researchers to add / supplement with additional biomarkers as needed. 

2- What specific mechanisms do the authors suggest might link elevated levels of CRP, INR, and fibrinogen to the development of NOAF in septic patients?

We thank the reviewer for this questions and highlight our discussion in which it is postulated that a raised CRP can increased calcium release in cardiac myocytes leading to depolarization and increased risk of arrhythmia. However a single unified pathophysiological method as to how CRP, INR and fibrinogen levels may lead to NOAF is lacking. Indeed it is unclear as to whether these are simple association and may reflect the general level of critical illness or whether these are causative. It is likely that NOAF occurs in a two hit hypothesis in which patients with a cardiac substrate that has undergone structural and electrical remodeling and is primed to develop AF undergoes a second hit that triggers the AF such as electrolyte imbalance or catecholamine release. Inflammation can lead to accelerate cardiac myocyte changes both structurally and electrically. It is possible that alterations in INR, fibrinogen and CRP are reflective of this inflammatory state such a during sepsis. 

3- In light of the findings, what recommendations do the authors provide for clinical practice regarding the monitoring and management of patients with both sepsis and NOAF?

Whilst specific recommendations for clinical practice are out with the scope of this review, we would highlight that NOAF is common and in particular is more common in these patients acutely unwell with septic shock. Similarly there appears to be a signal that higher CRP and WCC may increase the risk of NOAF. Therefore in those patients admitted with septic shock and high inflammatory markers clinicians should have a high index of suspicion that episode of tachycardia may signify NOAF that may have a significant impact on outcomes. 

4- The language and grammar of the manuscript could be improved to enhance the clarity and readability of the text. For instance, in the abstract of this manuscript:

... Medline, Cochrane Library and Scopus ... --- > ... Medline, Cochrane Library, and Scopus ...

We have reviewed the manuscript and improved the language and grammar.

---

## [Editor Report · Decision Letter 1]

15 Jan 2025

A Systematic Review on the Influence of Coagulopathy and Immune activation on New Onset Atrial Fibrillation in Patients with Sepsis

PONE-D-24-15357R1

Dear Dr. Johnston,

We’re pleased to inform you that your manuscript has been judged scientifically suitable for publication and will be formally accepted for publication once it meets all outstanding technical requirements.

Kind regards,

Chiara Lazzeri

Academic Editor

PLOS ONE
---

## [Editor Report · Acceptance letter]

19 Jan 2025

PONE-D-24-15357R1 

PLOS ONE

Dear Dr. Johnston, 

I'm pleased to inform you that your manuscript has been deemed suitable for publication in PLOS ONE. Congratulations! Your manuscript is now being handed over to our production team.

Kind regards, 

on behalf of

Dr. Chiara Lazzeri 

Academic Editor

PLOS ONE